# New Light in Polymer Science: Photoinduced Reversible Addition-Fragmentation Chain Transfer Polymerization (PET-RAFT) as Innovative Strategy for the Synthesis of Advanced Materials

**DOI:** 10.3390/polym13071119

**Published:** 2021-04-01

**Authors:** Valentina Bellotti, Roberto Simonutti

**Affiliations:** Department of Materials Science, Università Degli Studi di Milano-Bicocca, Via R. Cozzi, 55, 20125 Milan, Italy; v.bellotti1@campus.unimib.it

**Keywords:** PET-RAFT, controlled polymerization, photochemistry, advanced materials

## Abstract

Photochemistry has attracted great interest in the last decades in the field of polymer and material science for the synthesis of innovative materials. The merging of photochemistry and reversible-deactivation radical polymerizations (RDRP) provides good reaction control and can simplify elaborate reaction protocols. These advantages open the doors to multidisciplinary fields going from composite materials to bio-applications. Photoinduced Electron/Energy Transfer Reversible Addition-Fragmentation Chain-Transfer (PET-RAFT) polymerization, proposed for the first time in 2014, presents significant advantages compared to other photochemical techniques in terms of applicability, cost, and sustainability. This review has the aim of providing to the readers the basic knowledge of PET-RAFT polymerization and explores the new possibilities that this innovative technique offers in terms of industrial applications, new materials production, and green conditions.

## 1. Introduction

The introduction of reversible-deactivation radical polymerizations (RDRP) to the scientific community has provided relevant improvements over the classical radical polymerization technique thanks to the “living character” of this method. RDRP are based on the equilibrium between (macro) propagating radicals and dormant species through an activation/deactivation equilibrium [1]. The living chains spend most of their polymerization time in a dormant state and this allows to reach low dispersity (Đ), higher control over the molecular weight, and the possibility to continue the polymerization with fresh monomers in order to obtain complex architectures [2] (i.e., block, star, branched or hyper-branched network). RDRP is a collective term that embraces processes such as nitroxide-mediate polymerization (NMP) [3], atom transfer radical polymerization (ATRP) [4], and reversible addition-fragmentation chain-transfer polymerization (RAFT). In particular, RAFT follows a degenerative transfer mechanism in which there is no change in the overall number of radicals during the polymerization and it requires an external source of radicals, typically a radical initiator.

In recent years, many concerns have arisen about industrial development and environmental impact of controlled polymerization techniques, reflecting the necessity of more sustainable chemical processes. In this field, increasing attention has been paid to the development of light initiated controlled polymerization techniques, culminating in the introduction of light induced RAFT polymerization in 2014. 

Within the present review, our aim is to provide simple and representative examples of PET-RAFT polymerization, starting from the basic steps which brought to the development of this innovative technique. Attention has been posed on the reaction mechanism, the most relevant photoredox catalysts, the experimental setup that guarantee the green conditions provided by this technique. Among those, oxygen tolerance, catalyst recyclability and the use of light itself as trigger are pivotal features of this protocol. Finally, we decided to explore the most innovative applications offered by PET-RAFT in terms of industrial and medical fields such as the fabrication of advanced materials, design of 2D patterns for lithography, 3D printing, the synthesis of complex architecture structures endowed with antimicrobial activity or antifouling properties.

## 2. Form RAFT to PET-RAFT Polymerization

### 2.1. Reversible Addition-Fragmentation Chain-Transfer Polymerization (RAFT)

RAFT polymerization can be defined as one the most versatile of RDRP methods allowing the synthesis of complex architecture for a wide variety of functional monomers in different reaction conditions (aqueous or organic solution, bulk, dispersion, emulsion, miniemulsion) [5,6]. This technique was first developed at the Commonwealth Scientific and Industrial Research Organisation (CSIRO), an Australian Government agency responsible for scientific research, and announced in 1998 [7]. The living free-radical character was addressed by an available and simple class of organic reagents able to reversibly react with the propagating (macro)radical species by a chain transfer mechanism, generating the dormant specie. Such compounds are thiocarbonylthio-containing molecules (Z–C(=S)S–R), also called RAFT agents or CTA (chain transfer agent). The overall process (Figure 1) can be viewed as the insertion of monomer units between the S–R bond of the RAFT agent. The importance of R and Z groups nature is further discussed in Section 2.2.1.

The ability to switch between the propagating and dormant state makes negligible the termination reactions, thus it is the key factor that enables to obtain good control over the molecular weight, very narrow dispersity, and defined molecular architectures. These properties were once thought impossible to obtain via free radical processes, in which the length of the chain could not be well predicted or controlled. The key features of the well-known RAFT polymerization mechanism [5,8,9] is the sequence of addition-fragmentation equilibrium shown in Figure 2:

The first step is a common initiation reaction, the same that occurs in free radical polymerization. Thermal initiators (such as azobisisobutyronitrile, AIBN) have been the most widely used in RAFT polymerization and have the aim of producing the first radical oligomeric species. These later react with the thiocarbonylthio compound [(Z–C(=S)S–R (**1**)] in the initialization step, the addition depends on the reactivity of the C=S bond in comparison with the reactivity of the propagating radical ($P_{n}^{•}$). An equilibrium is established where the intermediate radical (**2**) fragmentation provides a polymeric thiocarbonylthio compound [(P_n_–C(=S)S–Z (**3**)] which is the dormant species, and a new radical ($R^{•}$) which acts as initiator and reacts with few monomers, ending up with a new propagating radical ($P_{m}^{•}$). A rapid equilibrium between the propagating radicals ($P_{n}^{•}$ and $P_{m}^{•}$) and the dormant end-capped state (**5**) by the way of intermediate (**4**) is settled and provides an equal probability for all chains to grow, obtaining low dispersity polymers. Due to this rapid equilibrium the concentration of radicals ($P_{n}^{•}$ and $P_{m}^{•}$) is lower than the more stabilize intermediate (**4**) and radical-radical termination reactions are suppressed but not completely eliminated, termination by combination or disproportion still occurs. For all these states to be true the equilibrium must be rapid or in other terms $k_{addP}\gg k_{p}$ and $k_{\beta }\gg k_{p}$. The equilibration of chains respects the polymerization rate is the key aspect to control over molar mass and dispersity. The number of dead polymer chains depends only on the number of radicals generated in the first step and includes both chains with an initiator fragment and the RAFT R-group at the α-end. This is a great advantage since the livingness of the system can be easily optimized by tuning the concentration of the initiator. At the completion of the RAFT polymerization most of the chains are still living, which means that at the ω-end there is the thiocarbonylthio group. This end-capped homopolymer can be seen as a macro RAFT agent and can be used to re-initiate the polymerization with a second different monomer, thereby achieving complex architectures such as block copolymers [10], star, branched, surface grafting or nanoparticles [11,12,13,14].

#### RAFT Agent

A very wide variety of RAFT agents with different structures suitable for particulars kind of monomers can be found in various reviews [15,16]. Effectiveness of the chain transfer agent, and so the accomplishment of the polymerization, depends on the monomer and the properties of free radical leaving groups R and Z. In fact, a key aspect in the choice of the more suitable RAFT agent is that the reactivity of C=S bond must be higher than the reactivity of C=C bond. This feature can be achieved by carefully selecting Z and R groups [15]. The Z group influences the stabilization of the intermediates (**2**) and (**4**), and activate or deactivate the C=S bond to monomer addition, which means that modify both the addition rate of the propagating radical $P_{n}^{•}$ to the RAFT agent and the fragmentation rate. In case of trithiocarbonates (Z = S-alkyl) or dithiobenzoates (Z = Ph) the radical is more stable and the C=S bond more reactive favoring the addition of more-activated monomers (i.e., (meth)acrylates and (meth)acrylamides, butadiene, isoprene, acrylonitrile, styrene). On the other hand, in xanthates (Z = O-alkyl) and dithiocarbamates (Z = N-alkyl) the lone pare of oxygen and nitrogen destabilize the intermediate radical and reduces the C=S double-bond character making the radical addition more difficult. For these reasons they are more suitable for les activated monomers (i.e., vinyl acetates, N-vinylpyrrolidone, N-vinylcaprolactam). R group influences the radical addition to the CTA by controlling the C=S reactivity in a similar way as the Z group. Furthermore, the R group nature is important for radical intermediate fragmentation, as it has to be a good leaving group. $R^{•}$ becomes the new initiator of the polymerization [5] and it has to ensure that all the chains initiate in the same time frame to obtain low dispersity index ($k_{i}>k_{p}$). A fine balance between radical stability and steric effects must be reached.

### 2.2. Photoinduced Electron/Energy Transfer (PET)-RAFT Polymerization

In the last years, great efforts have been made to develop controlled polymerization techniques triggered by external stimuli thanks to their ability to switch between ON and OFF states. Since the pioneering work of Otsu et al. [17] on the use of thiocarbonylthio compound as photoiniferters (where inifert stands for initiator–transfer agent–terminator [18]) and the development of RAFT living polymerization, photoactivation of the CTA has been the subject of much investigations. First studies, reporting direct photoactivation of the chain transfer agent using γ-radiation [19] turned out to be successful for polymer synthesis. However, the mechanism exploited a non-specific radical generation, which leads to chain imperfections and irreversible termination. Direct CTA activation under UV light has been also analyzed and extensively reviewed [20], this process is possible thanks to the already known weak *n* to *π** absorption of some thiocarbonylthio compound in the UV-Vis spectrum. The thiocarbonylthio compounds behave as photoiniferters materials, as proposed by Otsu, which upon exposure to light serve as the means of initiation, transfer, and termination. Even if no initiators or catalysts are needed following this protocol, the high energetic electromagnetic irradiation results in the loss of end-group fidelity due to the photolysis of the RAFT agent under UV light, meaning the loss of “living” character. For example, Davis and co-workers [21] attempted the polymerization of styrene using 1-phenylethyl phenyldithioacetate (1-PEPDTA) as the transfer agent under ultraviolet radiation at 365 nm. The polymerization was shown to proceed with living characteristics at low monomer conversion, but a significant broadening of the molecular weight distribution was observed at long irradiation times, due to the decomposition of transfer sites at the end of the polymer chain. Indeed, high energetic UV light has been used for removing the RAFT group by thiocarbonylthio-end degradation in post-modification protocols [22]. For these reasons over the last years, there has been a tendency to move toward higher wavelength radiation in the visible portion of the spectrum to reduce side reactions. In 2014 an innovative work of Boyer at al. [23] converges the development of visible light photoredox catalysis and the photoiniferter properties of CTA which culminated in PET-RAFT technology. This photoinduced electron/energy transfer-reversible addition-fragmentation polymerization is based on an alternative activation of the thiocarbonylthio compounds through photoredox catalysts (PCs) and results in several advantages with respect to the common RAFT protocol. First of all the avoidance of the initiation step, which can give unwanted α termini (via direct initiation) or ω termini (via termination of the growing chain by the exogenous radical) [24], employing an exogenous radical source (the photocatalyst). Moreover, the PC avoid the reaction heating, allowing low temperature polymerization in the presence of oxygen. Finally, the use of light as external stimulus imparts spatiotemporal control due to the possibility of switching ON and OFF the external source and provides orthogonality with other controlled polymerization techniques.

#### 2.2.1. PET-RAFT Mechanism

Notwithstanding RAFT mechanism is well understood, the interaction between the excited photoredox catalyst (PC*) and the RAFT agent has not yet been well investigated. Two are nowadays the proposed mechanism [25,26] as shown in Figure 3. The originally suggested path involves a photoinduced electron transfer from the excited state photoredox catalyst to the RAFT agent which becomes a thiocarbonylthio anion (this is possible owing to the strong reduction potential of the photoredox species used in PET-RAFT polymerization). Fragmentation of the last forms the initiating species ($R^{•}$ or $P_{n}^{•}$). The radicals may subsequently interact with the reduced RAFT agent and the oxidized PC respectively generating a dormant macro-RAFT species, closing the catalytic cycle. An alternative mechanism follows a photoinduced energy transfer pathway in which energy is transferred through an electron exchange mechanism generating an electronically excited RAFT species which undergoes fragmentation to generate propagating radicals as well as CTA radicals, which in the end recombine to produce a dormant polymer chain. This mechanism is presumed to follow a Dexter electron exchange pathway (P3T, photoinduced triplet energy transfer) in which the electron is transferred from the excited state donor to the ground state acceptor, which simultaneously transfers a ground-state electron back to the donor species, resulting in no net gain or loss of electrons for the molecules involved. The exchange of electrons does however result in the transfer of energy.

Recent computational studies on Ir and Ru complexes seem to suggest that triplet energy transfer through a Dexter electron exchange is more probable using Tris(2-phenylpyridine)iridium(III) (*fac-*Ir(ppy)_3_, Ir^(III)^) [27] and Tris(bipyridine)ruthenium(II) (Ru(bpy)_3_^2+^) [28]. The authors conclude that absorption of a photon in the visible region causes electronic excitation of the photocatalyst, populating higher energy singlet states, with subsequent rapid intersystem crossing populating the lowest excited triplet state of the photocatalyst. The mechanism presented for these two PCs cannot be generalized for all the other photocatalysts which need further investigation. In particular, some new studies underline that for porphyrin-based catalysts [29] electron/charge transfer, rather than energy transfer, is the likely operational mechanism for the PET-RAFT processes. Computational modeling has the great potential of assisting the further understanding of PET-RAFT mechanism for the wide range of photocatalyst employed but, so far, the research carried out is still limited.

#### 2.2.2. Photoredox Catalysts

The introduction of photoredox catalysts to initiate the polymerization has different advantages with respect to thermal initiated RAFT. Electromagnetic radiation guarantees spatiotemporal control and polymerization rate can be tuned by modulation of light intensity and wavelength. Furthermore, the reaction can be performed at room temperature and in presence of molecular oxygen (see Section 3.3). Finally, a very low amount of catalyst is needed (ppm range) in comparison with common thermal initiators concentration.

The first catalyst to be investigated by Boyer and coworkers was an iridium complex (*fac-*Ir(ppy)_3_, Ir ^(III)^) [23] which was employed with a wide variety of monomers, both conjugated and non-conjugated (i.e., (meth)acrylates, styrene, (meth)acrylamides, vinyl esters). Syntheses of block copolymers at room temperature in the presence of oxygen, were achieved with a light source operating at λ*_max_*= 435 nm (blue light) with 1–4.8 W power. Four RAFT transfer agents were investigated, 4-Cyano-4-(phenylcarbonothioylthio)pentanoic acid (CPADB), 2-(n-butyltrithiocarbonate)-propionic acid (BTPA), 3-benzylsulfanyl-thiocarbonylthiosulfanyl propionic acid (BSTP), and xanthate, since their higher redox potential with respect to Ir ^(IV)^/Ir ^(III)*^ couple. The mechanism was hypothesized to follow a photoelectron transfer. The results confirmed the success of this controlled polymerization techniques for both monomer families with low dispersity, good monomer conversion, an agreement between experimental, and theoretical M_n,_ and high end-group fidelity further confirmed by chain extension for block copolymers synthesis. PET-RAFT system was transferred in subsequent work in aqueous media through the use of a water-soluble ruthenium catalyst (Ru(bpy)_3_Cl_2_, Ru^(II)^) [30]. Different solvents were analyzed such as DMSO, acetonitrile, methanol, toluene, and water for the polymerization of *N,N’*-dimethylacrylamide (DMA) discovering that in water the rate of polymerization was slower than expected (k_p,app_ increase for high dielectric solvents) probably due to the solvation of the PC which decreases the catalytic activity. Nevertheless, living character (end group fidelity close to 100%) and excellent control of the molecular weight were achieved demonstrating also that the polymerization can be activated and deactivated by light (in “ON” and “OFF” environment). In the following years, a wide variety of photocatalysts for PET-RAFT polymerization was studied (Figure 4) including naturally derived photoactive compounds or metal-free organic catalysts moving toward greener polymer manufacturing.

Porphyrins and metalloporphyrins are important functional molecules with a large, conjugated ring structure and typically strong absorbance in the visible region of the electromagnetic spectrum and they exist in nature in the form of chlorophyll, heme, and VB_12_. Chlorophyll *a* (Chl *a*) [31] was the first porphyrin-based structure analyzed as photocatalysts and it was able to activate PET-RAFT polymerization. Shortly after also bacteriochlorophyll *a* (BChl *a*) [34] was investigated thanks to its strong absorption at long wavelengths from red to near-infrared (NIR) allowing to use of less energetic radiation with deep light penetration. Those initial works on chlorophyll paved the way to studies on a variety of non-toxic and low-cost metalloporphyrins with non-precious metals (Figure 4g). Zinc porphyrins [33] (in particular, 5,10,15,20-tetraphenyl-21H,23H-porphine zinc (ZnTPP)) were able to selectively activate photoinduced electron transfer-RAFT polymerization of trithiocarbonate compounds for the polymerization of styrene, (meth)acrylates and (meth)acrylamides under a broad range of wavelengths (from 435 to 655 nm). This photoredox catalyst has some advantages over the transition metal complexes above mentioned such as a lower inhibition period in presence of oxygen, likely due to a specific interaction between Zn and trithiocarbonate sulfur atoms as it happens in nature, and light absorbance at lower energy. Also, metal-free porphyrin (tetraphenylporphyrin (TPP)) [35] was able to activate the polymerization but with less efficiency. For this reason, covalent conjugation with CTA was used to create acceptor-donor molecules in which TPP acted as a light-harvesting antenna donating the excited electron to the electron-accepting thiocarbonylthio group. The proximity of reaction centers increasing the efficiency of electron transfer. Organic dyes provide another metal-free option for PET-RAFT polymerization, for this purpose eosin Y (EY), fluorescein, Rhodamine 6G, Nile red [32] have been tested, together with 10-phenylphenothiazine (PHT) [36] but only EY and PTH have proved effective. Finally, inorganic semiconducting metal oxides, including TiO_2_ [37] and ZnO [38], have also been employed as photocatalysts for PET-RAFT polymerization. Titanium dioxide is particularly interesting since widely used as photocatalysts for oxygen reduction reactions and solar cells, it is also cheap, chemically stable, and with low toxicity; but nowadays only two examples are reported in the literature in terms of PET-RAFT polymerization [37,39]. The photoredox catalysts herein described are just a part of the wide classes of PCs suitable for PET-RAFT polymerization. Less common photocatalysts that have been used for PET-RAFT purposes are Perovskite nanocrystals (CsPbBr_3_) [40], CdSe quantum dots (QDs) [41,42], tertiary amines (tris(2-(dimethylamino) ethyl)amine, *N*,*N*,*N*′,*N*′′,*N*′′-pentamethyldiethylenetriamine, and triethylamine) [43], and graphitic carbon nitride [44]. In Table 1 the main advantages and disadvantages of the most used catalysts are reported.

#### 2.2.3. Compatibility with Other Controlled Polymerization Techniques

PET-RAFT polymerization, following the same pathway of conventional RAFT, presents the same compatibility with other polymerization reaction conditions, but introduces more attractive attributes due to an additional level of control. In fact, some examples of combined polymerizations, one-pot and sequential PET-RAFT/ring-opening, PET-RAFT/ATRP have been reported in the literature.

Ring-opening polymerization (ROP) with an anionic or a cationic chain-growth mechanism is a powerful tool for synthesizing biodegradable polyesters. It has been demonstrated that PET-RAFT technique can be combined with ROP in order to synthesize block copolymer in one-pot strategy [45]. In fact, PCL-*b*-PMA copolymer has been successfully synthesized by either sequential or simultaneous diphenyl phosphate (DPP)-catalyzed ROP and iridium-catalyzed PET-RAFT polymerization without interference and sacrificing chain livingness. Well-defined PCL-*b*-PMA block copolymers with low molecular weight distributions (Đ < 1.15) were successfully prepared. In addition, the characteristic temporal control of PET-RAFT gave rise to independent behavior during the simultaneous polymerization. This features has been exploited by Fu et al. [46], which combined a visible light sensitive photoacid as catalyst to trigger ROP in the one-pot polymerization with ZnTPP catalyzed PET-RAFT. Polymerization has been switched between two different monomers by just changing light wavelengths affording orthogonal reactions. With a photoredox catalyst responsive to red light (635 nm) and a merocyanine-based photoacid activated under blue light irradiation (460 nm), (photo)-ROP and PET-RAFT polymerizations can proceed independently, with both block lengths being controllable. Furthermore, thermal ROP was combined with PET-RAFT in a ingenious way [47] taking advantage of light and heat as orthogonal external stimuli for living and controlled polymerization. When temperature was risen up to 50 °C anionic ring opening polymerization (AROP) catalyzed by quaternary onium salts was induced, whereas irradiation with blue light activate the Ir(ppy)_3_-catalyzed PET-RAFT polymerization. Since AROP of the 2-(phenoxymethyl) thiirane (POMT) does not occur at room temperature and PET-RAFT of *N,N*-dimethylacrylamide (DMAm) did not occur in the absence of light ; by carefully tuning the two external stimuli multiblock copolymers could be synthesized in a one-pot strategy.

Finally, recent work by Theriot, Boyer et al. [48] employed *N,N*-Diaryl dihydrophenazines as organic photoredox catalyst for PET-RAFT and organocatalyzed atom transfer radical polymerization (O-ATRP) sequential block co-polymerization of PMA-*b*-PMMA. Given the similarity of the role of the PC as an electron-transfer agent in both PET-RAFT and O-ATRP, the two techniques were combined, starting from the synthesis of a dual initiator (EtBriB–BTPA), which contained a PET-RAFT initiating trithiocarbonate moiety and an O-ATRP initiating alkyl bromide moiety attached to the R-group side. One key difference between the two polymerization types is the amount of PC required to produce polymers with low Đ; typically, O-ATRP requires a higher PC concentration (500 ppm), whereas PET-RAFT requires a significantly lower PC concentration (10 ppm). For this reason, PET-RAFT polymerization was the first to be performed and then the PC concentration was increased to initiate O-ATRP, ending up with the desired block copolymer. Selectivity between monomers was achieved by the rational choice of the chain transfer agent since the PMMA block is expected to polymerize only via O-ATRP, as the BTPA moiety is not able to polymerize methacrylates.

## 3. PET-RAFT Polymerization: A Greener Approach

Electromagnetic radiation guarantees spatiotemporal control, whereas the modulation of light intensity and wavelength tunes the polymerization rate. Light allows moving toward mild and green reaction conditions such as room temperature polymerization even in the presence of oxygen [49], avoidance of exogenous radical sources which can give unwanted α or ω termini [24] and orthogonality with other techniques [45,46,47,48]. All these important features result in several implemented applications (analysed in the applications section) which go from the synthesis of 3D composite materials to biological applications.

### 3.1. Light Sources

The selection of the light source is crucial since both wavelength and intensity are critical parameters in photochemical reactions. Wavelength is related to photons energy thus to the electrons transition at higher energy orbitals, whereas light intensity influences the amount of photon (photon flux) that interact with the photocatalyst [50]. Both these factors affect the polymerization rate and can be effortless modified providing additional control of the chemical process. In principle, all the external light sources on the market are suitable for mediating photochemical reactions, including xenon, fluorescent lamps, mercury, halogen, LEDs, or even sunlight. Among them, light-emitting diodes (LEDs) have become dominant in photochemistry in the new millennium thanks to numerous advantages (Figure 5). LEDs are based on electroluminescent emission, leading to a very precise emission wavelength with a narrow half width [51], which is pivotal for efficient polymerization. Very low voltage is required to power the lamp, which brings to less consumption of electricity, and less heat produced by the single compact LED units which may eliminate the need of heatsink or more complex equipment [52]. Finally, the cost and very long lifetime compared with the other light sources make them very appealing in photochemical processes.

Temporal control is ensured for photopolymerizations: by simply turning the light “ON” and “OFF”, the reaction can be instantaneous stopped and started (Figure 6). In particular, for PET-RAFT polymerization the removal of light triggers the switching of the RAFT agent in the dormant state as demonstrated in the pioneering work of Boyer et al. [53]. This remarkable ability is unreachable for thermally and chemical-induced polymerizations, thus limiting their applicability.

Furthermore, the employment of light allows performing localized polymerization. The spatial control is owed to the activation of the polymerization only in specific areas enabling the one-pot synthesis of 3D materials with very specific and customizable properties. This aspect is critical to enhancing techniques in which interfaces play an important role, such as lithography, and will be further discussed below.

### 3.2. Recyclable Photocatalysts

Heterogeneous catalysis can be a valuable tool for the development of environmentally friendly PET-RAFT polymerization. The presence of photoredox catalyst in the polymeric matrix, even if in very low concentration (ppm range), can cause issues for the final material in terms of toxicity or side reactions, but also leads to polymer degradation. In fact, over the past few years removal and recycle of the PC has become a hot topic for the scientific community (Figure 7). For example, the use of catalysts anchored to solid support rather than in solution guarantees easy removal and recycle. Some relevant examples can be found in literature such as the use of celluloses as the supporting materials for the immobilization of ZnTPP [54,55] using simple grafting chemistry. PET-RAFT polymerization was effectively regulated, and the novel heterogeneous catalyst was restored by simply squeezing and washing the material after each polymerization cycle. The catalytic performance was maintained for at least three cycles. Furthermore, silica nanoparticles were modified with EY on the surface through covalent bonds to prepare a novel heterogeneous catalytic system for PET-RAFT polymerization [56]. Mesoporous silica nanoparticle conjugated to Eosin Y (EY-SNPs) were recovered via centrifugation at the end of the polymerization and they were reused to perform multiple polymerization cycles. Moreover, Johnson and coworker [57] developed a thermally responsive photoredox catalyst gel covalently bound to 10-phenylphenothiazine (PTH) which was able to activate the polymerization with the aid of both temperature and light. The Gel-PTH behaves like a heterogeneous catalyst so it was easily removed and used directly in another polymerization, up to six photo-reactions. Not only anchorage on a solid support can guarantee the full recovery of the PCs but also intrinsic heterogeneous systems such as nanoparticles catalyst have proven useful tools in recycling. For instance, magnetic ferrite nanoparticles have been used for Recyclable-catalyst-aided, Opened-to-air, and Sunlight-photolyzed (ROS)-RAFT polymerizations [58]. The catalyst was recovered and reused five times for the polymerization of MMA and the control over the polymerization was preserved. Finally, in a very recent work of Weiss et al. [59], semiconductor quantum dots have proven to be good recyclable photocatalyst for aqueous controlled polymerization of polyacrylamides and polyacrylates. Amicon^®^ Ultra centrifugal spin filter usually used for protein separation was effectively employed for the purification of QDs from both the polymer and the unreacted monomer. The absorbance spectra confirmed the successful separation and QDs were used for further polymerization reactions. This method allows for maintaining higher colloidal stability compared to immobilization to fibrous or nanoparticles.

### 3.3. Oxygen Tolerance and High Throughput Polymerization

Usually, in radical polymerization, to avoid termination reactions caused by the presence of oxygen, the reaction mixture is degassed (freeze-pump-thaw cycles) in sealed reactors and polymerization is conducted under an inert atmosphere (nitrogen or argon). Even if those methods are effective, they require time, inherently expensive gases, and complex equipment. PET-RAFT polymerization is a valuable alternative that can solve the problem in terms of applicability, cost, and sustainability. It is well-known, since the establishment of free radical polymerization, that oxygen is a radical scavenger, inhibiting radical polymerization by reacting with propagating radical to form peroxy species and thus terminating the polymerization. Tolerance to oxygen is an intrinsic feature of the PET mechanism [60] and it is attributed to the process represented in Figure 8: (i) conversion of the molecular oxygen into singlet oxygen by intermolecular triplet-triplet annihilation (TTA) of energy transfer from the excited photoredox catalyst to molecular oxygen and (ii) reaction of singlet oxygen with the solvents (such as DMSO) or other added reducing agents. A wide range of singlet oxygen scavengers are known in literature [61] but up to now only tertiary amines, ascorbic acid, anthracene, limonene have been successfully used in open-air PET-RAFT polymerization [62]. The consumption of oxygen rate is different depending on the photocatalyst of choice. For instance, Ir and Ru-based photocatalyst [30] need an induction period of several hours to consume oxygen and start the polymerization whereas ZnTPP no induction period was required [33]. This strongly depends on photocatalyst quantum yield to transform triplet oxygen into singlet oxygen which in the case of porphyrin is >70%.

Other systems have been developed and proven effective in the consumption of molecular oxygen from the reaction environment such as enzyme degassing [33,63]. Glucose oxidase (GOx) is an inexpensive enzyme that in the presence of glucose leads to the reduction of molecular oxygen into a quenching species such as hydrogen peroxide. This strategy has been successfully applied to PET-RAFT polymerization [64], replacing the more common quenching agents, but its use is limited since it does not exploit the intrinsic capability of the photocatalyst and it makes the reaction mixture more complex.

The oxygen tolerance feature has been recently exploited to permit high throughput synthesis of controlled polymer libraries at low reaction volumes (40 μL plates) [65]. The creation of polymer libraries can be used as a systematic approach for the optimization of polymerization reaction conditions as well as the rapid generation of complex phase diagrams, reducing the waste of monomers and solvents. This method is particularly valuable in the design of sophisticated polymer architectures (i.e., multiblock copolymers, star polymers, bottlebrushes and so on). Moreover, the possibility to perform several reactions in parallel, it allows to synthetize large sets of polymers varying each single parameter relevant for the reaction, simplifying the physico-chemical polymer analysis and characterization. High throughput approach, by PET-RAFT oxygen-tolerant polymerization, was evaluated for a broad range of monomers (acrylates, methacrylates, acrylamides, and styrenic monomers) which were polymerized directly in multiwell plate by employing ZnTPP as photocatalyst [66]. ZnTPP is not water-soluble, for this reason, DMSO was chosen as solvent, which also can quench singlet oxygen generated by photosensitization, and the polymerization proceeded under yellow LED light (λ = 560 nm, 9.7 mW/cm^2^) affording optimal results in terms of monomer conversion and Đ in presence of oxygen. Furthermore, Boyer et al. [67] have exploited ZnTPP as self-reporting photocatalyst, able to both mediate the polymerization and act as high throughput online monitor for monomer conversion. The polymerization was conducted in 384-well microtiter plates and the change in fluorescence properties of the photocatalyst was used as an indicator of reaction progress avoiding sampling of the reaction mixture. Other interesting and more specialized studies on screening activity have been carried out, for instance, Chapman and co-workers [66] exploit this method to investigate the effect of polymer structure on protein binding (lectin *concanavalin*A, *Con*A). This study concerned the used of ZnTPP, for the polymerization of a range of acrylamides in low volume at high monomer conversion (Figure 9). Knowing that glycopolymers bind to lectins, a library of mannose-functionalized linear, 3-arm and 4-arm star glycopolymers were synthetized by post-polymerization modification. NHS functional groups were incorporated in the polymer chain and used to attach a strained alkyne (dibenzocyclooctyne–amine, DBCO-NH_2_). Performing Huisgen cycloaddition between the alkyne and an acetylated azido-mannose the glycopolymer was generated and screened in order to study the binding affinity to *Con*A.

Other oxygen quenchers and photocatalysts have been exploited beyond ZnTTP in ultralow volume photopolymerization. For example [49], eosin Y in the presence of ascorbic acid and green light irradiation (530 nm) initiates the polymerization of acrylamides, acrylates, and methacrylates in a 20 μL microtiter plate without prior deoxygenation. This method was effective for the synthesis of complex structures such as diblock copolymer, star polymers, and polymeric nanoparticles via Polymerization-Induced Self-Assembly (PISA) approach. This latter consists in chain-extension of a solvophilic homopolymer with a solvophobic block, inducing, concomitantly to the growth of a second insoluble block, the in situ self-assembly of the resulting diblock nano-object.

### 3.4. Waste Minimization

An important concept of green chemistry relies on atom economy, meaning the ability of minimizing by-products, thus the reaction waste. Low by-products formation is possible using selective techniques, a feature that can be granted by the employment of photoredox catalysts. In fact, the electron transfer from the PC during irradiation is selectively accepted by the chain transfer agent, initiating the controlled polymerization. Moreover, light sources as LED [51], with narrow half width and low energy consumption, are safe and the visible radiation avoid the side reaction characteristic of high-energy UV or γ radiations such as the degradation of the CTA [19,20]. The direct generation of radical using the PC avoids the addition of exogenous radical sources which can give unwanted α or ω termini [24], dead chains polymers that remains as impurity in the synthesis of more complex architectures. Finally, the photocatalyst amount used on PET-RAFT remains in the ppm range and, as cited before, PCs recyclability is an hot topic in recent years. All these are interesting features which follow the green chemistry guidelines minimizing by-products formation.

## 4. Applications

Polymer science is nowadays pivotal for everyday life, thus covers a large portion of the actual market. One of the main challenges in polymer science is the development of new advanced materials with precise properties for multidisciplinary fields such as biotechnology, medical science, formulation chemistry, composite materials, microelectronics, plastic solar cells, biosensors, packaging, and so on. In this regard, photochemistry is a promising tool for developing innovative strategies appealing to both academic and industrial environments. The combination of photochemistry and RDRP provides good reaction control and does simplify elaborate reaction protocols. PET-RAFT polymerization is a valuable alternative and can solve the problem in terms of applicability, cost, and sustainability. Besides, the process is fully scalable thanks to the similarity with the traditional radical polymerization. This section has the aim to provide the reader some of the main advantages of PET-RAFT polymerization, in terms of industrial and biomedical applications, enhancing the production process or opening the doors to applications once thought impossible for radical polymerization.

### 4.1. Fabrication of Advanced Materials for Commercial Applications

#### 4.1.1. Manufacturing of Nanocomposites

The term nanocomposite refers to heterogeneous materials of nanometer dimension whose surface (biocompatibility, wettability, chemical reactivity, and adhesion) and physicochemical (thermal stability, elasticity, permeability, and so on) properties can be finely tuned [68]. Surface grafting of polymeric chains on substrates is a promising technique for the preparation of new advanced materials with enhanced properties [69,70]. For instance, polymers grafts are employed to protect substrates from light damage or corrosion, impart substrate with flame retardancy, prevent nanoparticle aggregation, or to modify surface properties of inorganic fillers to increase affinity for polymeric matrix, making easier to process for industrial applications. Three are the most exploited protocols for the synthesis of inorganic-polymeric materials: (i) grafting to, (ii) grafting from, and (iii) grafting through whose mechanisms are already fully reviewed [71,72]. RAFT polymerization has been already applied for surface modification of several inorganic compounds, few examples are gold [73], iron oxide [74], silica [75], and titanium dioxide [76] nanoparticles. For instance, our group [77] developed TiO_2_@PS nanocomposite in a fashionable controlled way through grafting-from polymerization, but only a little research is available on PET-RAFT.

Surface-living polymerization triggered by light provides a new effective method for the tuning of thickness and density of the polymer brushes on the surface in open to air vessels, eliminating the time-consuming deoxygenation process and avoiding the use of complex equipment and expensive inert gases. For instance, SiO_2_@poly(NIPAM-AcFl) thermoresponsive nanocomposite, in which N-isopropylacrylamide (NIPAM) and acrylated fluorescein were grafted on silica nanoparticles, was manufactured at room temperature in presence of oxygen and with no metal contamination thank to the use of surface-initiated PET-RAFT technique [78]. Interestingly, in this work the photoredoxcatalyst of choice was fluorescein that was acrylated to play two roles during the process, both the photocatalyst and the monomer to form a self-catalytic copolymer with NIPAM. Afterward, additional work on silica-containing nanomaterial has been published [79]. The authors exploit temporal control upon light regulation to produce SiO_2_ nanoparticles coated with well-control poly(*N,N*-dimethylacrylamide)brushes of target molecular weight and high grafting density (Figure 10). A grafting-from approached was selected by anchoring PDP chain transfer agent with the subsequent polymerization of DMA at 25 °C under blue LED light irradiation (4.8 W, λ*_max_* = 465 nm). The versatility and robustness of this method were demonstrated by its extendibility to other monomers and silica-assisted nanomaterials such as Pt@SiO_2_ core-shell and Fe_3_O_4_@SiO_2_ Janus nanoparticles.

Other interesting and very recent works consist of the use of quantum dots (QDs) as both photocatalyst and substrate for the grafting procedure. Quantum dots are semiconductor particles with a dimension of few nanometers and strong visible light harvesting properties, which have been proven ideal candidates for photocatalytic purposes. Egap and coworkers [42] selected cadmium selenide (CdSe) quantum dots as the photocatalyst and were able to create QDs-PMMA nanocomposite material in one-pot polymerization. Moreover, Silicon Quantum Dots (SiQDs) [80] with amino end-functionality have been anchored to silicon wafer substrate exploiting electrostatic interaction and further used as photocatalyst for surface-initiated PET-RAFT polymerization, as shown in Figure 11. The role of SiQDs is therefore to connect polymer with the substrate along with photocatalytic purpose. CTA with carboxylic end-functionality was grafted on NH_2_-SiQDs and polymerizations were performed under blue LED (6 W, λ*_max_* = 460 nm, 2 mW cm^−2^) irradiation at room temperature. Temporal control of the polymerization technique has been demonstrated by switching the light source ON and OFF. When the light was ON, a linear increase in molecular weight was observed whereas polymerization stop in the OFF stages. The versatility of this synthetic design allows using the SiQD-coated wafer as an efficient recyclable photocatalyst for PET-RAFT polymerization with the aid of an electrode clip to clamp the silicon wafer. The latter could be easily immersed and pulled out from the solution.

#### 4.1.2. Design of 2D Patterns for Lithography

The design of functional interfaces is a key factor for a wide range of industrial applications, which goes from optoelectronic devices to solar cells and biological active surfaces, and it is already been addressed in the past decades by reversible deactivation radical polymerization techniques such as ATRP and RAFT [81]. Besides temporal control, the early employment of light allows performing localized polymerization on various substrates. The activation of specific areas, exploiting photomasks, guarantees the one-pot synthesis of 3D materials with customizable properties. This advancement is granted to the low life time of both CTAs and PCs so that the radical diffusion cannot be higher than few nanometers [82,83]. Surface initiated (SI)-photopolymerization spatiotemporal ability has been already used for the generation of 2D surface patterning.

Hawker and coworkers have previously developed SI-ATRP based on visible-light mediation for the fabrication of patterned polymer brushes on surfaces [84,85,86]. For instance, they reported an inspirational work [87] in 2016 describing a solution-exchange photolithography protocol based on the combination of stopped-flow techniques and reduction photolithography, which is able to afford hierarchically structured substrates using photo ATRP under an inert atmosphere. Even if photo-ATPR has been more extensively studied in those years for surface pattern generation, PET-RAFT polymerization can bring several improvements to this field thanks to the milder reaction conditions, such as low temperature, oxygen, and functional group tolerance. In fact, very recently, Pester and coworker [88] demonstrated the ability of PET-RAFT to target specific film thickness thanks to the characteristic reaction control with high chain-end fidelity on silica surfaces by grafting the CTAs through Z-group approach. Polymerization proceeded in open to air vessels using both ZnTPP and Ir(ppy)_3_ as photocatalysts under blue and yellow light (405 and 590 nm) with an equal rate comparing to inert atmosphere. Advanced 3D topographical complex patterns were achieved using an optical lens array, able to project the pattern of a grayscale image on the CTA-substrate, and SI-PET-RAFT polymerization was performed. In Figure 12 an optical micrograph of the final surface is reported, the shades of gray are consistent with different levels of photon flux in separated areas, meaning light intensity. Finally, compatibility between photoinduced and traditional RAFT was addressed by subsequently reaction to tethered block copolymer architectures via extension process.

Moreover, Seo et al. [89] exploited metal-free SI-PET-RAFT polymerization in order to fabricate 3D polymer brushes under aqueous conditions, using the photomask technology. Eosin Y was the photocatalyst of choice under green light (λ = 530 nm) and the oxygen tolerance was addressed by in situ enzyme degassing approach, using glucose oxidase (GOx). As expected, brush thickness linearly increased within irradiation time for a wide variety of water-soluble functional monomers (NIPAM, DEA, HEA, PEGMEMA, SA, DMAEMA) only in the light irradiated areas. Patterning experiments were also performed on diblock copolymers as shown in Figure 13. An initial uniform layer was polymerized, followed by chain extension with different monomers through a photomask, high spatial fidelity of copolymer brushes was achieved by this facile and versatile method.

Those earlier studies highlight the great potential of SI-PET-RAFT protocol for the growth of finely tuned and hierarchically organized interfaces for lithography purposes. We are confident that in the foreseeable future more attention will be paid to spatial-controlled surface-initiated photopolymerization such as SI-PET-RAFT due to its numerous advantages, enhancing the field of lithography.

#### 4.1.3. New Advances in 3D Printing

An outstanding application of photoinduced polymerization is photomediated 3D printing. 3D technology allows the reduction of waste through additive manufacturing and the formation of complex geometries. Besides, photoinduced methods such as stereolithography (SLA), digital light processing (DLP), and projection micro stereolithography (PμSL) offer higher resolution, faster printing, and versatility compared to thermal techniques [90]. Besides, room temperature polymerization is pivotal for the incorporation of thermosensitive compounds in composite material design [91]. The low oxygen tolerance and slow polymerization rate of traditional RDRP techniques preclude their application in this field but the introduction of PET-RAFT technology overcomes their main limitations, providing additional opportunities for the synthesis of advanced customizable materials [90,92]. Rapid and oxygen tolerant polymerization process under environmentally friendly conditions are the key factors for the success of PET-RAFT technique together with the possibility of finely tuning the molecular weight, thus controlling the mechanical properties.

The first efforts in the enhancement of photomediated 3D printing using PET-RAFT polymerization have been proposed for the first time in the past year. Jianyong Jin and coworkers [93] successfully exploited PET-RAFT for a photocurable resin formulation which was used in an open-to-air digital light processing 3D printer under mild conditions (Figure 14). Eosin Y was irradiated under blue (λ = 483 nm, 4.16 mW/cm^2^) or green (λ = 532 nm, 0.48 mW/cm^2^) light in the presence of triethylamine reducing agent that granted the oxygen tolerance. Each layer was exposed to light for 5 min in order to photopolymerize, reaching a maximum build speed of 730 μm/h.

Noteworthily, when free radical polymerization is used the 3D printed final material is immutable, in terms of further monomer insertion. On the other hand, post-printing is achievable with PET-RAFT, thanks to the living functional thiocarbonate group that terminates the polymer chains. Furthermore, one of the advantages of light is that wavelength and intensity can be used to manipulate the material meanwhile it is manufactured or at the end of the process. These features combined with the living character of the polymerization and the spatiotemporal control were used for the synthesis of smart materials, able to respond to environmental stimulus changing their structure over time, entering into the concept of 4D printing [94,95]. Boyer and coworkers [96] exploited the intrinsic tunable properties of light for the one-pot synthesis of sensitive materials with spatially tuned properties. Erythrosin B was selected as photocatalyst together with tertiary amine co-catalyst for the copolymerization of DMAm and PEGDA achieving a water-soluble polymeric network with the aid of a DPL 3D printer. Studies at different light exposure in terms of time and intensity revealed the alteration of mechanical properties of the final material such as the storage modulus (*E’*) and glass transition temperature (*T*_g_), hence 4D materials with spatially resolved properties were targeted. The polymeric network was printed using a decreasing amount of light for each layer and was then dehydrated and re-swelled for actuation. As shown in Figure 15b the printed object starts to deform consequently with the swelling toward the less light-exposed region. When it was removed from the water and dried, it flattened and then inverted the curvature consequently to complete dehydration. In another interesting work, the living behavior of PET-RAFT together with its spatial control were exploited for the growth-induced bending of a preprinted strip [97]. In this paper the authors relied on the above-mentioned features, performing a chain extension experiment on the already printed dormant object, achieving a more complex 4D structure. In particular, Eosin Y, BTPA, poly(ethylene glycol)diacrylate, and *N,N*-dimethylacrylamide were copolymerized layer by layer under blue light (λ = 405 nm) for 30 s in open air thanks to the presence of triethanolamine. The resulting printed object was a polymeric strip (120 × 20 × 0.6 mm) with living end functionalities. This latter was immersed in a growth medium containing the photocatalyst, the tertiary amine, and a second fresh monomer (butyl acrylate, BA) for the chain extension experiment. After 30 min of irradiation under green light, a significant bending of the strip was achieved (Figure 15a). The bending occurs only in the spatial region of irradiation, whereas the remaining part of the strip (which stayed in the dark) was unchanged in shape. The final qualitative data show an outstanding preliminary result of 4D responsive material with a completely novel method.

More efforts can be done hereafter to improve PET-RAFT protocol in 3D printing fields. Nevertheless, the reported examples provided a significant step toward the enhancement of 3D and 4D technology opening the doors to new developments.

#### 4.1.4. New Industrial Possibilities

In summary, the combination of photochemistry with controlled living radical polymerization such RAFT is a promising tool for developing innovative strategies in materials production. Fabrication of nanocomposites through surface-grafting is an already established method in materials science. Even though, surface-living polymerization triggered by light provides a new effective method for tuning thickness and density of the polymer brushes in open to air vessels, eliminating the time-consuming deoxygenation process and avoiding the use of complex equipment and expensive inert gases. Moreover, the spatial control of PET-RAFT allows the activation of specific areas, a pivotal feature in lithography. The combination of photoinitiated polymerization and photomask technology guarantees the one-pot synthesis of 3D materials with customizable properties, finding applications in optoelectronic devices, solar cells and biological active surfaces. Finally, an outstanding new application of PET-RAFT relies on photomediated 3D printing in which rapid and oxygen tolerant polymerization process under environmentally friendly conditions are key factors of its success.

### 4.2. PET-RAFT for Medical Purposes

The intrinsic features of PET-RAFT approach, such as room temperature polymerization in the presence of molecular oxygen and with very low catalyst concentration, allow the polymerization to proceed in physiological conditions, making this technique suitable for biological applications.

#### 4.2.1. Compatibility with Biological Moieties: Bioconjugation

Since the very beginning of PET-RAFT development, its future key role in biological and medical science was highlighted by chain extension of water-soluble macro-CTAs with a biological moiety such as bovine serum albumin (BSA) [30]. This latter retained its biological activity during the polymerization, opening the doors to further studies for the combination of synthetic polymers with biological entities for synthesis of polymer–drug conjugate therapeutics. Some interesting bioactive species in this regard can be peptides, proteins, nucleotides and lipids.

In fact. biomolecule-polymer conjugates offer the possibility of combining the advantageous properties of both biological macromolecules and synthetic polymers in a single macromolecular architecture. For instance, poly(ethylene glycol) (PEG) has been extensively studied in therapeutics to form biomolecule-polymer conjugates thanks to its ability to increase the in vivo half-life, protecting the therapeutic form immune system recognition (stealth property) and its biodegradability [98]. The development of controlled polymerization techniques such as NMP, ATRP, ROP, and RAFT introduced the ability to prepare tailored polymers more suitable for biomedical application. Bio-conjugation can be accessed through three different routes [99] which are the same described for nanocomposite materials: “grafting-to” in which the polymer is first synthesized and then coupled with the biomolecule, “grafting-through” in which monomers functionalized with a specific payload are polymerized, and “grafting-from” in which the biological moiety is functionalized with a reactive molecule to start the polymerization. This last is the most used since offer many advantages in terms of graft density and purification. One of the main limitations of grafting-from techniques is that the common polymerization conditions are not suited to the stability of the biomolecule but are compatible with the PET-RAFT protocol. Protein-polymer bioconjugate via PET-RAFT mechanism using BSA [100] (Figure 16), lipase [101] or lysozyme [102] have been reported. Cysteine or lysine residues of proteins have been exploited to attach a suitable functionalized CTA through the R-group end and the polymerization was initiated by Ru complex or eosin Y water-soluble photocatalysts. The biological activity of each protein was retained and demonstrated to be close to the one of free protein.

In addition to proteins, other biological macromolecules such as lipids [103] and DNA [104] (Figure 17) were exploited to synthesized polymer bioconjugate via PET-RAFT grafting-from protocol. DNA-polymeric conjugates are usually achieved by grafting-to approach exploiting phoshoramidite coupling chemistry which requires organic solvents, strong basic condition, and several hours. On the other hand, the authors exploited PET-RAFT chemistry anchoring two different CTAs (BTPA and CPADB) for the grafting-from polymerization of (meth)acrylic monomers under blue LED light, in water, at room temperature. The resulted amphiphilic conjugates can form supramolecular architecture such as micelles with appealing drug delivery application.

Moving from biomolecules such as proteins or nucleic acids to living cells, oxygen tolerance and low polymerization temperature become strict requirements to achieve conjugation, since freeze-pump-thaw cycles cannot be used with live cells. Recently, Hawker and co-workers [105] reported the use of mild EY/triethanolamine surface-initiated PET-RAFT polymerization to grow polymer brushes directly from the surface of living yeast cells. Polymerization occurred in pH 7.4 phosphate-buffered saline (PBS) at low monomer concentration, in 300 μL reaction volumes, and at room temperature within 5 min. Fast polymerization is crucial to avoid injury to the cells. This strategy enables chain growth to be initiated directly from chain-transfer agents anchored on the surface of living cells using either covalent attachment or non-covalent insertion. In the initial strategy, the CTA was covalently attached to the amino groups of yeast cell surface proteins through a suitable bifunctional linker and the polymerization occur under the LED 465 nm irradiation. The non-covalent attachment was instead achieved by using a lipid analog bearing a RAFT head group, a hydrophilic spacer, and a non-charged DPPE-mimicking C16 tail. This last was able to non-covalent insert into human Jurkat cell membranes, used as a model. Notably, the authors concluded that cell aggregation and assembly could be controlled by graft synthetic polymer on the surface.

#### 4.2.2. Antifouling Properties

The term biofouling refers to the adhesion and accumulation of biological matter such as proteins, microorganisms, seaweed, or even small animals on surfaces [106] (Figure 18). This phenomenon is undesirable for many applications, especially in medical and marine fields. The design of advanced antifouling materials, providing resistance to nonspecific interactions with cells or proteins, is crucial for the development of new technologies specifically targeted to avoid biofouling. Medical biofouling concerns the interactions of exogenous tools such as implants and medical equipment with complex biological fluids [107,108]. In this field the developing of antifouling materials is pivotal since the nonspecific interaction can reduce the circulation time of nanocarriers, cause a nonspecific response of affinity-based biosensors, the blockage of porous membranes, and bacterial attachment on contact lenses [109]. Furthermore, biofouling can be also associate with marine environment where micro- and macro-organisms appears on ships and underwater structures inducing corrosion, fuel consumption and frequent maintenance [110].

Nature offers inspiration in this respect by means of low adhesion, wettability, micro texture behavior, which can be mimicked for commercial purposes. Antifouling coating characterized by polymeric brushes on the surface is one of the main tools capable of preventing fouling of materials in contact with biological matter maintaining low toxicity of the final commercial product.

Since the development of PET-RAFT technique, few studies have been reported in literature on the synthesis of antifouling materials. For instance, Baggerman and coworkers [111] in a very recent work synthetized antifouling polymer brushes on silicon surfaces exploiting LED light (λ = 410 nm, 2.9 W) and eosin Y as metal-free catalyst. The antifouling surface was created in four steps starting from a naked silicon surface exploiting a grafting-from approach where the CTA was anchored through the R functionality (Figure 19). Polymerization of oligo(ethylene glycol) methacrylate (MeOEGMA), N-(2-hydroxypropyl)methacrylamide (HPMA), and carboxybetaine methacrylamide (CBMA) proceeded in oxygen-containing environment. Different thicknesses were obtained in the range of 4 to 45 nm as well as complex three-dimensional structures thanks to the controlled light-triggered nature of PET-RAFT polymerization. The formation of 3D structured polymer brush layers was achieved with a patterning mask, which allows tuning the thickness in the exposed regions.

Moreover, several recent works were found to exploit PET-RAFT surface-initiated polymerization on PVA hydrogel membrane. PVA is biocompatible and it is currently used for many biological applications such as biosensors, artificial cartilage, or cornea. However, common complications of the implanted materials such as cells absorption of protein via nonspecific interactions limit their applicability. Zwitterionic monomers such as carboxybetaine methacrylamide (CBMA) [112] and [2-(methacryloyloxy) ethyl]dimethyl-(3-sulfopropyl) ammonium hydroxide (MEDSAH) [113] have been grafted onto PVA hydrogel membrane via SI-PET-RAFT polymerization under visible light and in the presence of oxygen. In both cases, high grafting density where achieved, up to 50-60%, and the final material was effective in both anti-protein adsorption and anti-cell capacity (>50%) with no cytotoxicity. Finally, Zhongkuan Luo and co-workers [114] grafted glycidyl methacrylate (GMA) and 2-hydroxypropyltrimethyl ammonium chloride chitosan (HACC) on the surface of PVA hydrogel for artificial cornea production. This work is the demonstration of the orthogonality of PET-RAFT polymerization with post-modification reactions. Firstly, GMA was grafted on the hydrogel surface exploiting (Ru(bpy)_3_Cl_2_) as photocatalyst, and triethylamine (TEA) for the oxygen tolerance. This monomer is very popular to produce functional materials thanks to the epoxy functionality, which can directly react with amine, alcohols, and carboxylic acids to achieve post-modification functionalization. In fact, the authors decided to exploit the epoxy chemistry for the anchorage of HACC via ring opening reaction. The antifouling properties of this material increased due to the steric repulsion of p(GMA-HACC) polymer brush.

#### 4.2.3. Antimicrobial Activity

Microbial cells can adhere to any surface starting to proliferate building up a biofilm. The latter ensures cell survival, even under severe environmental conditions, and makes them less susceptible to biocides. Antimicrobial surfaces provide an optimal solution to the above-mentioned issues preventing the growth of biofilms or killing the bacterial in proximity to the surface. Antimicrobial polymers are an alternative option to disinfectants (hypochlorite, quaternary ammonium salts, or alcohols) which are considered environmental pollutants and assist the development of resistant microbial strains. In fact, since the development of antimicrobial polymers, several reviews can be found in the literature to easily understand the state of the art [115,116,117,118]. Briefly, antimicrobial polymers are the synthetic alternative of antimicrobial peptides (AMPs). The rational design of those polymers consists in cationic amphiphilic (hydrophilic–hydrophobic) macromolecular structures able to target the bacterial cytoplasmic membrane. The polymer can be composed by a hydrophilic functional block bearing cationic charge and a hydrophobic block or can be random copolymers with the same chemical composition just described. The positive charges increase the binding affinity for bacterial whereas the lipophilic portion is able to insert into the cell membrane, causing damage of the structural organization and integrity of the membrane. The magnitude of degradation increases with the increasing of polymer length. The most studied charged groups for this purpose are quaternary ammonium salts (QAS), quaternary pyridinium or phosphonium salts.

The combination of high throughput synthesis with PET-RAFT polymerization allows the rapid generation of polymeric libraries and the screening activity of antimicrobial polymers against different microorganisms (Figure 20). This approach offers new insight into the understanding of the adhesion ability and bacteria specificity of the synthesized polymers.

Gaojian Chen group [119] reported a recyclable-catalyst aided sunlight-photolyzed RAFT polymerization carried out in multiwell plates. High throughput technology afforded the synthesis of glycopolymers composed of hydrophobic, cationic, and sugar repeating units with variable compositions; their antibacterial activity and cytotoxicity were easily tested. Gibson group [120], as well as Boyer and coworkers [121,122], demonstrated the rapid PET-RAFT synthesis of a library of antimicrobial polymers, using visible light irradiation, without the need for degassing or purification step. Boyer investigated the effect of monomer distribution within linear high-order quasi-block copolymers on their antimicrobial properties. Cationic:hydrophobic:hydrophilic monomers with a ratio of 50:30:20 were used and specific placement of these monomers in the polymer chain resulted in different antimicrobial activities since in this controlled way they can mimic the secondary structure of well-investigated antimicrobial peptides (AMPs). Both hydrophobic/hydrophilic variations and cationic monomer variation respect the original ratio resulted pivotal versus different microorganisms. These studies highlight the possibility of test multiblock copolymers made via a highly efficient polymerization technique against a range of bacteria in short time and systematically manner. Finally, in a very recent work [123] high throughput process and flow polymerization were merged for upscaled production of antimicrobial polymers. In this work the authors highlight the possibility to build up a library of antimicrobial polymers, testing their antimicrobial activity through structure-properties analysis via both plate and flow polymerization. Ternary copolymers featuring cationic, hydrophilic, and hydrophobic moieties were selected. The cationic polymer was a protected ter*-*butyl (2-acrylamidoethyl) carbamate (BOC-AEAm), which can be easily deprotected in acidic condition, and was fixed in terms of amount. The hydrophobic (DMAEA, PEA, EHA) and hydrophilic (HEA, PEGA, DEGA) monomers ratio was changed. The most promising polymers were upscaled in flow reactor leading to a production of optimized antibacterial polymer at a rate of 27.2 g/day. Interestingly, comparable properties were obtained both with small and large-scale production protocols.

**Figure 20 polymers-13-01119-f020:**
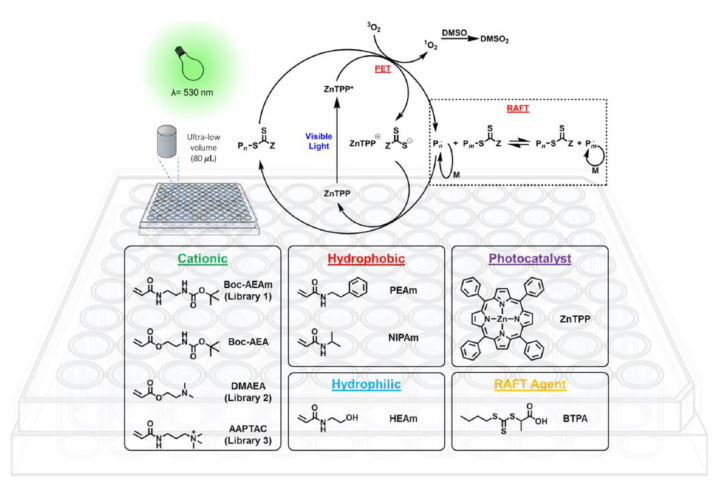
Schematic representation of PET-RAFT process. Reproduced with permission from ref. [121].

## 5. Conclusions and Outlook

In conclusion, the combination of controlled radical living polymerization such as RAFT with a great variety of photoredox catalyst culminating in PET-RAFT protocol offers valuable tools for the developing of innovative synthetic protocols in different fields. The living polymerization properties of RAFT are preserved, therefore low polydispersity well-defined polymers can be achieved, these are capable of being chain-extended producing more complex structures. Moreover, new features are introduced by the using of light such as spatial and temporal control, oxygen tolerance and room temperature polymerization. The latter are key factors which allow to satisfy the principles of green chemistry, moving toward environmentally friendly manufacturing both in academia and industry.

The photocatalysts can be natural derived or metal free compounds, potentially degradable or recyclable, and can be used in orthogonality with other polymerization techniques such as ROP and ATRP. Intrinsic selectivity of photopolymerization combined with very low amount (ppm range) of PC avoid unnecessary waste and by-product formation.

Oxygen tolerance and spatial control are incredible features for industrial applications. Oxygen tolerance is attractive in terms of time and cost since it makes the polymerization preparation faster without the requirement of further complex equipment as well as expensive gases. It allows to made compatible the radical polymerization with sensible systems such as biological macromolecules or even cells. In addition, oxygen tolerance has been able to merge polymer chemistry with high throughput synthesis and screening of polymeric libraries. On the other hand, spatial control is very appealing in the chemistry of interfaces allowing the synthesis of various novel materials with customizable properties, depending on the applications. This is true for nanoparticle decoration, lithography, preparation of antifouling and antimicrobial surfaces, as well as 3D and, more interestingly, 4D stimuli responsive printable materials.

We are just at the dawn of this remarkable technique, but the herein described examples demonstrate its incredible potential in many fields of application. Nevertheless, some important aspects still need to be considered in the immediate future to improve PET-RAFT polymerization.

Among these, more comprehensive computational studies are required to fully understand the catalytic-polymerization mechanism. This would provide the opportunity to easily select new suitable photocatalysts based on simple structure-activity relationship. In addition, considering that conventional RAFT is intensively used with flow reactor set up and the similarity between RAFT and PET-RAFT, there are many possibilities to develop PET-RAFT process in flow mode, although this process has not been extensive investigated yet.

Finally, in our opinion, a future opportunity to enhance this innovative polymerization technique could be the study of novel natural-derived monomers and solvents, which can move the next step toward greener manufacturing. In fact, the principles of green chemistry rely on the continues research of alternative and environmentally friendly reaction media, increasing the reaction rate, reducing the temperature, and optimizing the atom economy. For example, dihydrolevoglucosenone (Cyrene^TM^) [124] is a bio-based molecule, derived from cellulose, which can be used as dipolar aprotic solvent, similarly to DMF. In addition, methyl 5-(dimethylamino)-2-methyl-5-oxopentanoate (Rhodiasolv© PolarClean) [125] is another very recent non-toxic alternative to common polar aprotic solvents. Finally, ionic liquids are a new class of pure ionic compound, liquid at low temperature, which are promising replacement for organic polar solvents. Nevertheless, they have never been used in PET-RAFT polymerization, whereas classical radical polymerization exploited ionic liquids both as solvents [126,127] and monomers [128].

## Figures and Tables

**Figure 1 polymers-13-01119-f001:**
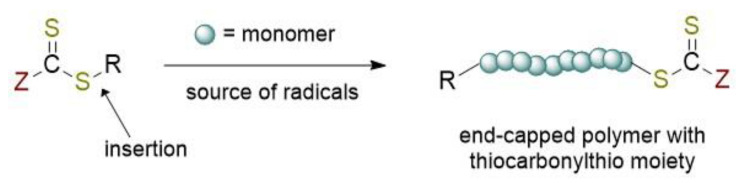
Chemical structure of the RAFT agent and the overall insertion product of the reversible-addition chain transfer polymerization.

**Figure 2 polymers-13-01119-f002:**
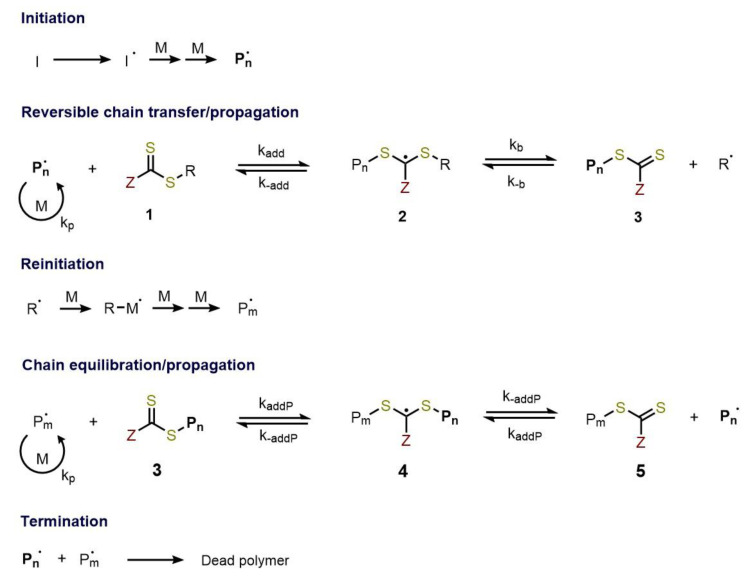
Mechanism of RAFT polymerization.

**Figure 3 polymers-13-01119-f003:**
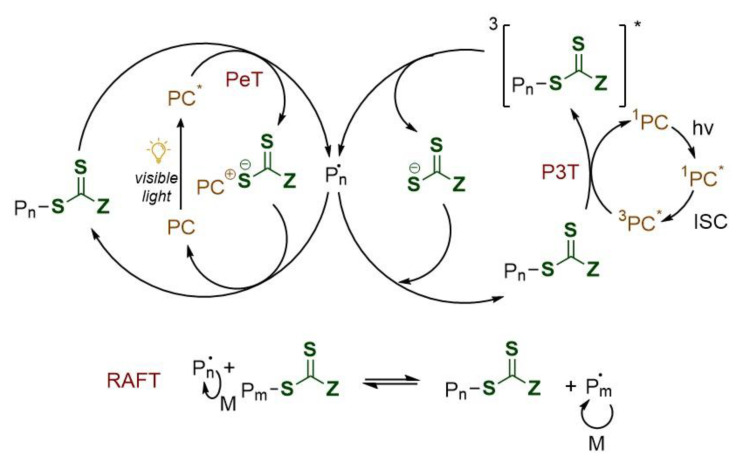
Proposed mechanisms for PeT (**left**) and P3T (**right**) RAFT polymerization. PC: photocatalyst; PeT: photoinduced electron transfer; P3T: photoinduced triplet energy transfer (Dexter electron exchange); ISC: intersystem crossing; *: excited state.

**Figure 4 polymers-13-01119-f004:**
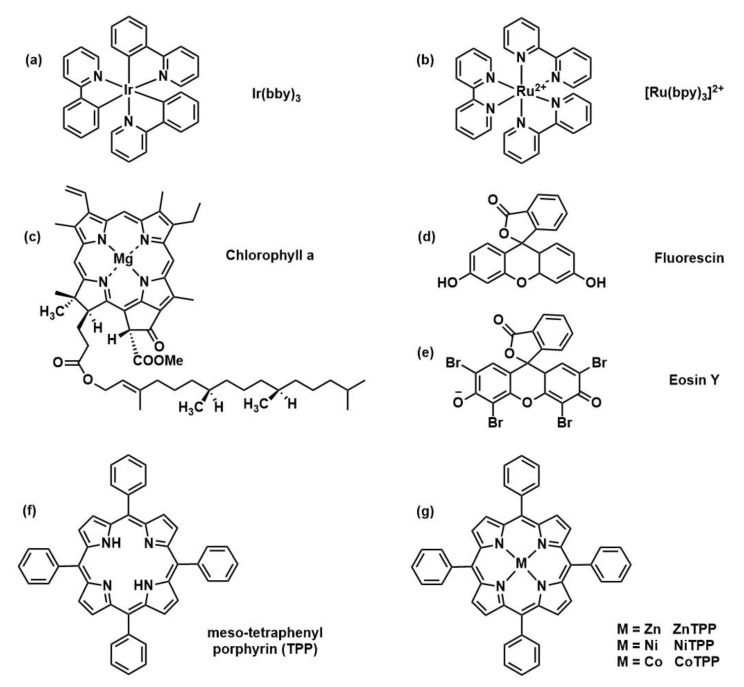
Selection of catalyst structures: (Ir(ppy)_3_) [23], (Ru(bpy)_3_Cl_2_) [30], chlorophyll *a* [31], Fluorescine [32], Eosine Y [32], and (metallo)porphyrin-based structures [33].

**Figure 5 polymers-13-01119-f005:**
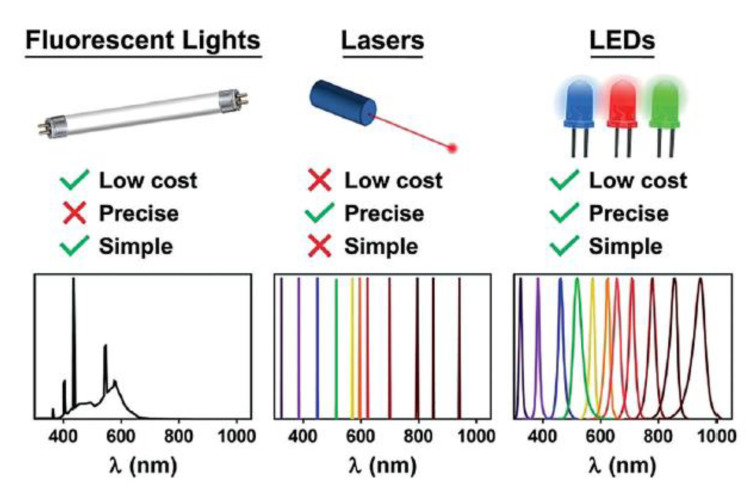
Main advantages of LED light sources. Reproduced with permission from ref. [50].

**Figure 6 polymers-13-01119-f006:**
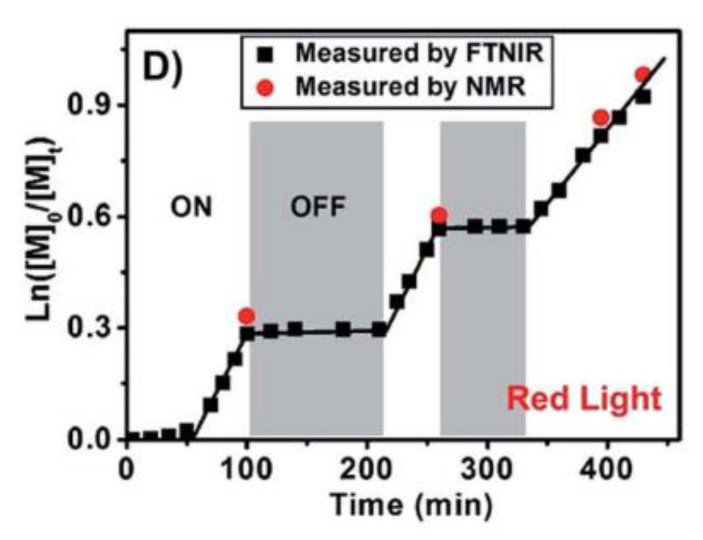
Temporal control of PET-RAFT technique. Reproduced with permission from ref. [31].

**Figure 7 polymers-13-01119-f007:**
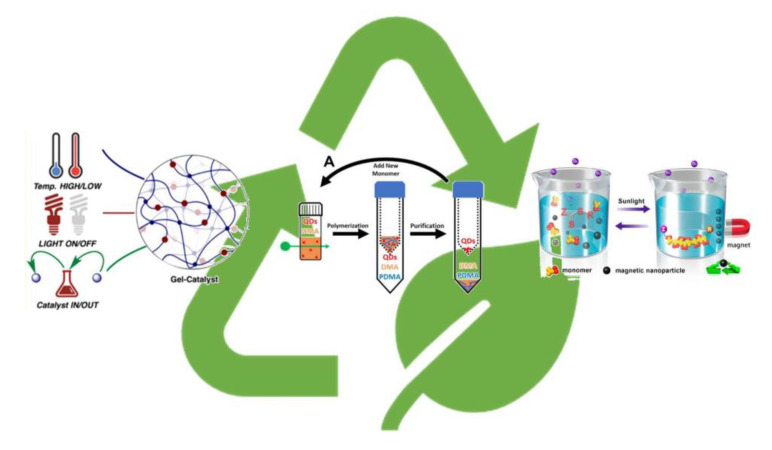
Recyclability of photoredox catalyst in PET-RAFT process. Adapted from refs. [57,58,59].

**Figure 8 polymers-13-01119-f008:**
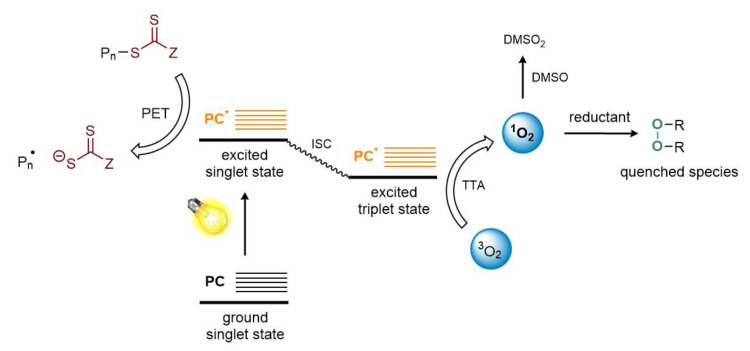
Schematic representation of the photophysic process involved in the intrinsic oxygen tolerance of PET-RAFT. ISC: intersystem crossing; *: excited state.

**Figure 9 polymers-13-01119-f009:**
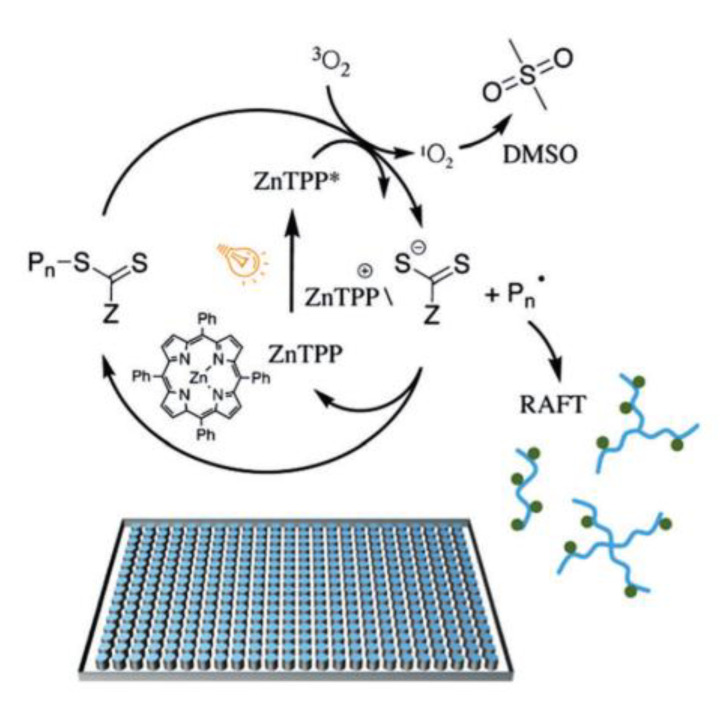
Ultralow volume PET-RAFT polymerization without any prior deoxygenation step and direct investigation on protein binding. Reproduced with permission from ref. [65]. *: excited state.

**Figure 10 polymers-13-01119-f010:**
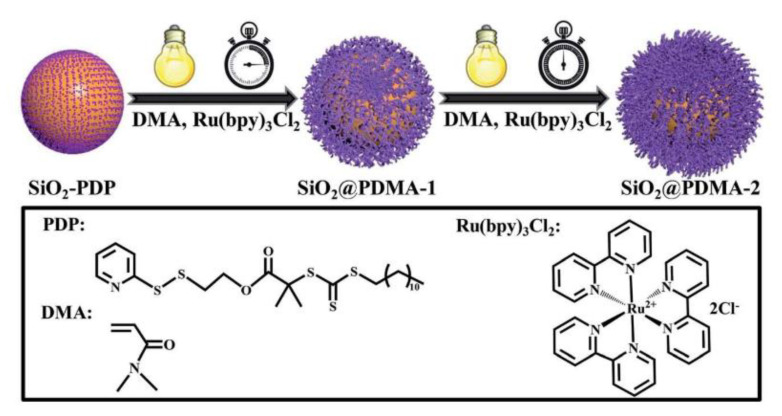
Synthesis of silica-polymer nanocomposites exploiting PET-RAFT approach. Reproduced with permission from ref. [79].

**Figure 11 polymers-13-01119-f011:**
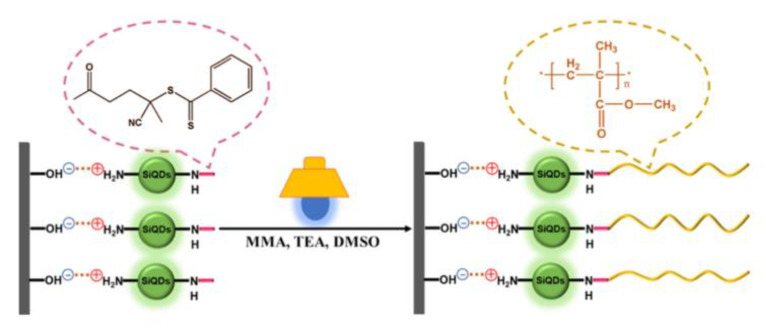
Reaction scheme of in Situ SiQD-Catalyzed Surface-Initiated PET-RAFT Polymerization. Reproduced with permission from ref. [80].

**Figure 12 polymers-13-01119-f012:**
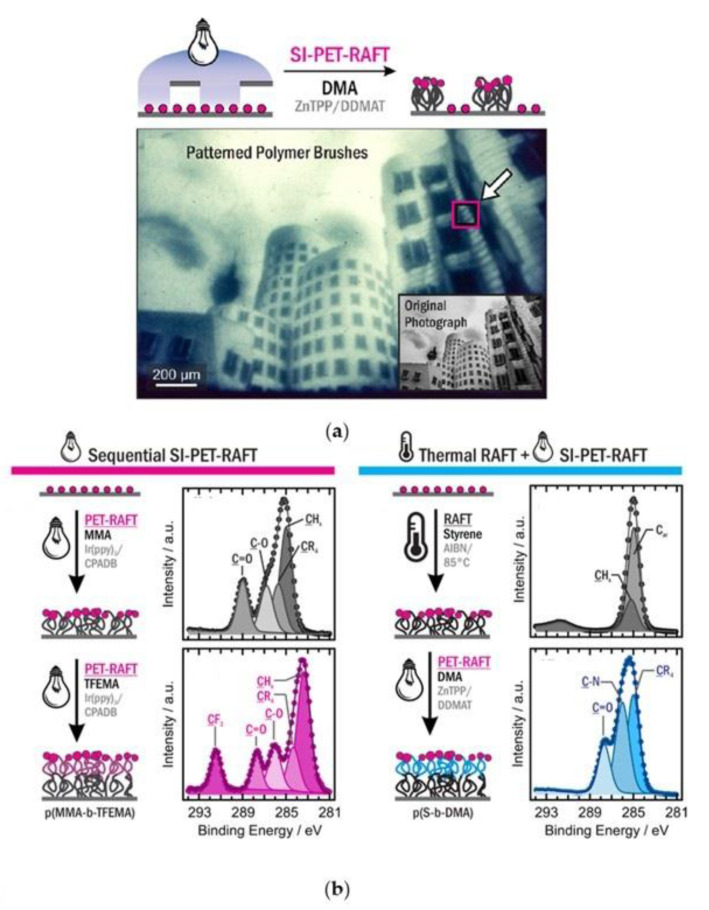
On the top (**a**): SI-PET-RAFT polymerization via localized irradiation of a CTA-functionalized substrate. Spatial control was achieved using a photomask on the SiO_2_ substrate via reduction photolithography process. On the bottom (**b**): a combination of thermal initiated and photoinitiated RAFT polymerization. Reproduced with permission from ref. [88].

**Figure 13 polymers-13-01119-f013:**
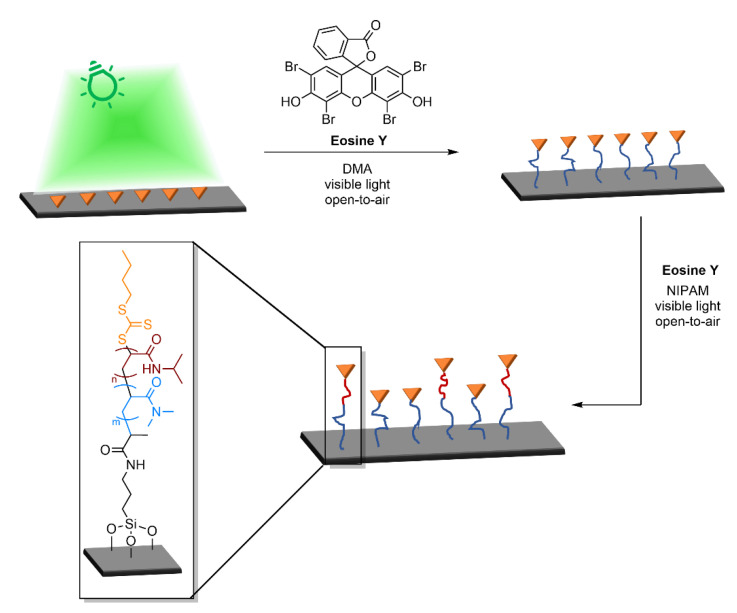
Schematic representation SI-PET-RAFT polymerization exploiting BTPA chain transfer agent anchored to a substrate. P(DMA-block-NIPAM) block copolymer synthesis exploiting photomask to obtain specific surface pattern.

**Figure 14 polymers-13-01119-f014:**
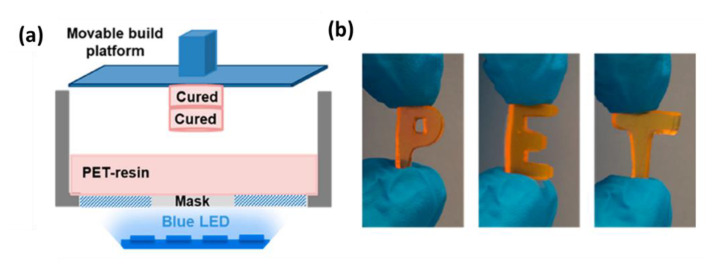
Schematic representation of (**a**) 3D printing process using a bottom-up DLP printer under blue LED light and (**b**) the optical images obtained by the printing process. Reproduced with permission from ref. [93].

**Figure 15 polymers-13-01119-f015:**
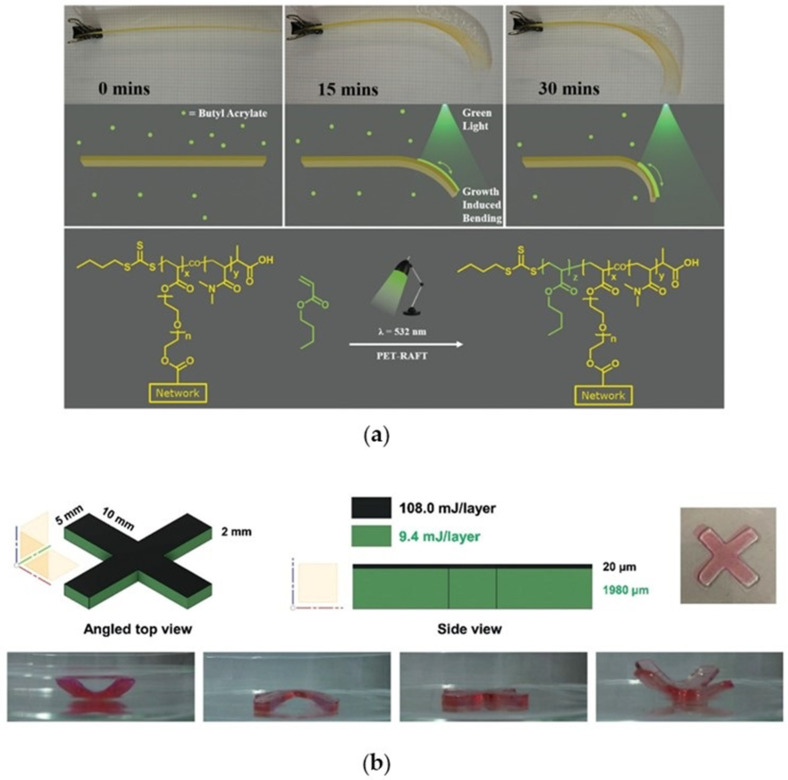
(**a**) Growth-induced bending process using PET-RAFT technique on a pre-printed strip. (**b**) Swelling induced actuation of spatially resolved 3D printed material in water. Reproduced with permission from ref. [96,97].

**Figure 16 polymers-13-01119-f016:**
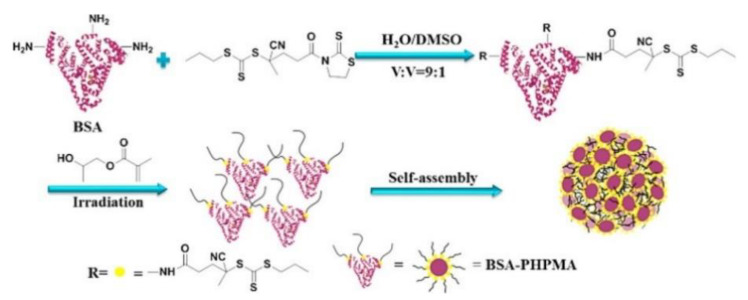
Synthesis of BSA Macro-CTAs and in-Situ Polymerization-Induced Self-Assembly of polymer biconjugate nanoparticles. Reproduced with permission from ref. [100].

**Figure 17 polymers-13-01119-f017:**
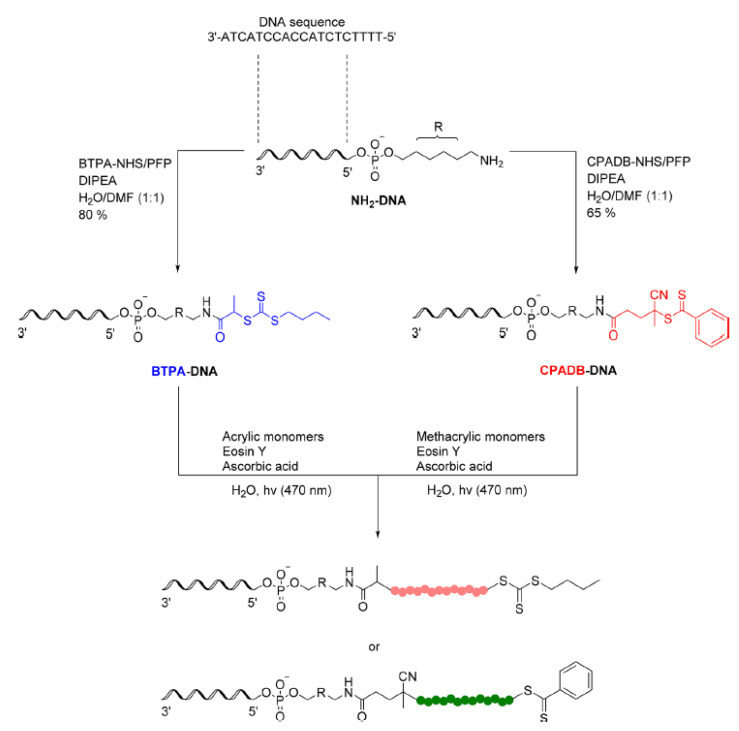
PET-RAFT grafting-from approach from DNA. Reproduced with permission from ref. [104].

**Figure 18 polymers-13-01119-f018:**
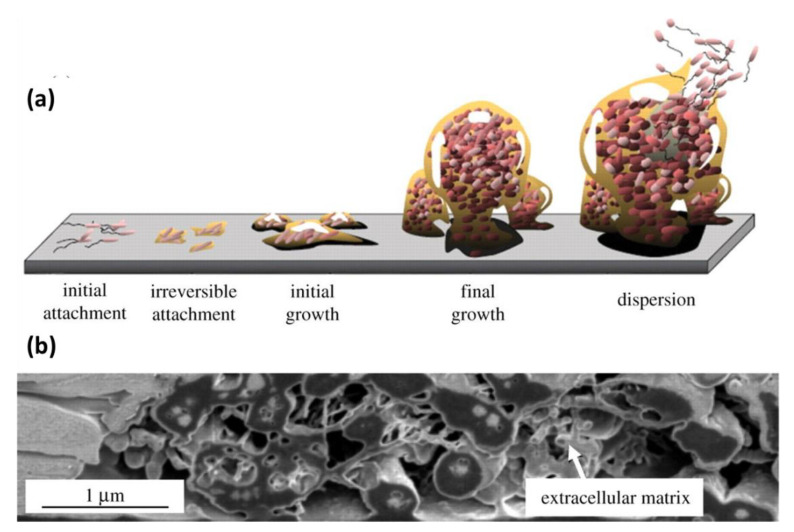
Biofilm development process. (**a**) Five-stage colonization process and (**b**) SEM of biofilm cross section highlighting the bacterial extracellular matrix morphology. Reproduced with permission from ref. [106].

**Figure 19 polymers-13-01119-f019:**
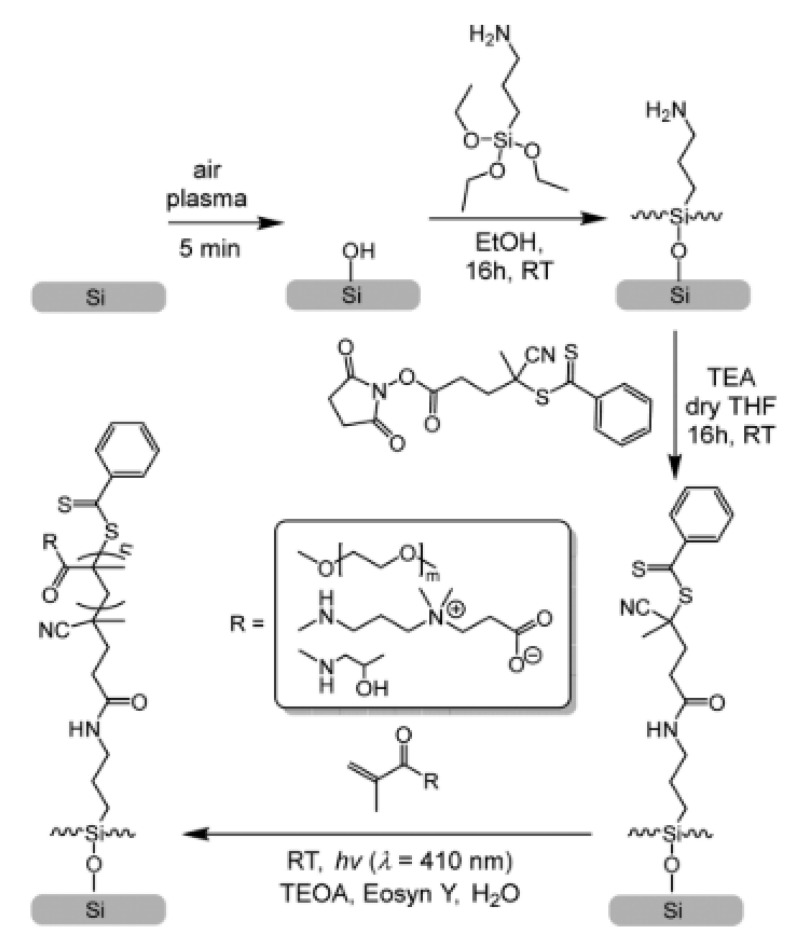
Reaction scheme to produce antifouling surfaces. Reproduced with permission from ref. [111].

**Table 1 polymers-13-01119-t001:** Main advantaged and disadvantages of the most representatives photoredox catalysts.

PC	Ref	Advantages	Disadvantages
(Ir(ppy)_3)_	[23]	Low energy absorption	Precious metal, expensive, soluble in few organic solvents
[Ru(bpy)_3_Cl_2_]	[30]	Solubility in wide variety of solvents (also polar)	Precious metal, expensive
Porphyrin-based	[31,33,34]	Non precious metal, low-cost, low-toxicity, multiple absorption peaks	Presence of impurity, effective only with trithiocarbonates
Eosine Y	[32]	Metal free, low cost	Sacrificial electron donor required, slow kinetics
Metal oxides (ZnO, TiO_2_)	[37,38]	Easy to recycle low-cost, low-toxicity	High energy irradiation
QDs	[41,42]	Easy to recycle, modulable absorption wavelength	Toxic compound
Tertiary amines	[43]	Metal free	High energy irradiation

## Data Availability

Not applicable.

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
