# Peer review of "New Light in Polymer Science: Photoinduced Reversible Addition-Fragmentation Chain Transfer Polymerization (PET-RAFT) as Innovative Strategy for the Synthesis of Advanced Materials"

_polymers, 2021, doi:10.3390/polym13071119_

Round 1

Reviewer 1 Report

The authors indented to present an extensive review article on Photoinduced Reversible Addition-Fragmentation Chain Transfer Polymerization with the aim of providing the basic knowledge of PET-RAFT polymerization and explore the new possibilities that this innovative technique offers in terms of industrial applications, new materials production, and green conditions. This version of the manuscript is informative and educational in most of the given examples; nevertheless, I would recommend this manuscript for publication after addressing the following issues:

  1. Importantly, the text is missing an introduction part which comprehensively place the aim of the review in a broad context and highlight why it is important.
  2. I would like to draw the attention of the authors that IUPAC has deprecated the use of the term polydispersity index, having replaced it with the term dispersity, accordingly please modify the text.
  3. In the photoredox catalyst section, it could be useful to present in tabular form the pros and cons of each catalyst system.
  4. The authors define the PET-RAFT process as greener approach; thus, they have outlined the importance of utilizing light as a trigger, catalyst recyclability, and oxygen tolerance, nevertheless, it should be also briefly summarised the concept of waste minimisation, particularly of by product formation, as the latter is directly corelated the environmental impacts of green chemical
  5. It is essential to add a section in which the compatibility of PAT-RAFT with other synthetic conditions is briefly discussed.
  6. From page 20 on, the numbering of the Figures is not correct, please double check.
  7. In similar manner, the numbering of Section “Conclusion” is not correct. Moreover, I would suggest to rename the last section as Conclusion and Outlook, thus to provide some overview of potential improvement of the process which eventually broaden the concept and toolbox of PET-RAFT. It is of similar importance also to discuss the limitations which the PET-RAFT process faces, explicitly in regard to its green chemistry features, and to mention how these limitations could be overcome in the future.
  8. Since it is aimed to provide a comprehensive review on the topic, it should be of crucial importance also to discuss how computational modelling could assist to understand the underlying catalytic mechanism and structure–property relationship of photocatalysts.
  9. I would suggest to summarize the new possibilities that this innovative technique offers in terms of industrial applications under a separate sub-section within the Main Section: 3 Applications.
  10. Last but not at least, in order to further improve the manuscript, I would like to suggest to improve the English; explicitly the text should be checked by a native English speaker before further submission.

Reviewer 2 Report

Dear Editor and Authors,

I include my revision in the manuscript file. Please read it carefully and revise according to my comments.

This is very good work provided by your team.
